# The Relationship between Retained Primitive Reflexes and Hemispheric Connectivity in Autism Spectrum Disorders

**DOI:** 10.3390/brainsci13081147

**Published:** 2023-07-30

**Authors:** Robert Melillo, Gerry Leisman, Calixto Machado, Yanin Machado-Ferrer, Mauricio Chinchilla-Acosta, Ty Melillo, Eli Carmeli

**Affiliations:** 1Movement and Cognition Laboratory, Department of Physical Therapy, University of Haifa, Haifa 3498838, Israel; 2Department of Neurology, University of the Medical Sciences of Havana, Havana 10400, Cuba; 3Department of Clinical Neurophysiology, Institute for Neurology and Neurosurgery, Havana 10400, Cuba; 4Northeast College of the Health Sciences, Seneca Falls, New York, NY 13148, USA

**Keywords:** retained primitive reflexes, autism spectrum disorders, maturational delay, neuronal synchrony, bottom-up processing, top-down processing

## Abstract

Background: Autism Spectrum Disorder (ASD) can be identified by a general tendency toward a reduction in the expression of low-band, widely dispersed integrative activities, which is made up for by an increase in localized, high-frequency, regionally dispersed activity. The study assessed ASD children and adults all possessing retained primitive reflexes (RPRs) compared with a control group that did not attempt to reduce or remove those RPRs and then examined the effects on qEEG and brain network connectivity. Methods: Analysis of qEEG spectral and functional connectivity was performed, to identify associations with the presence or absence of retained primitive reflexes (RPRs), before and after an intervention based on TENS unilateral stimulation. Results: The results point to abnormal lateralization in ASD, including long-range underconnectivity, a greater left-over-right qEEG functional connectivity ratio, and short-range overconnectivity in ASD. Conclusions: Clinical improvement and the absence of RPRs may be linked to variations in qEEG frequency bands and more optimized brain networks, resulting in more developmentally appropriate long-range connectivity links, primarily in the right hemisphere.

## 1. Introduction

Autism spectrum disorder (ASD) is a complex neurodevelopmental syndrome characterized clinically by language impairment, lessened social interaction, varied cognitive deficits, and behavioral stereotypes [1,2,3,4,5,6,7,8]. People with ASD may act, speak, interact, and learn differently than typical people [3,7]. The abilities of individuals with ASD can differ considerably [3]. Some individuals with ASD may have great conversational abilities, while others may be nonverbal [2,3,4,6]. Individuals with ASD need much help daily; others can work and live without support [3]. Although symptoms of ASD occasionally improve, they typically appear before the age of three and might continue for the rest of a person’s life [3,7]. Children can demonstrate ASD symptoms within the first 12 months of life [3]. Symptoms may not develop until 24 months of age or later. Up to about the age of 18 to 24 months, children with ASD may learn new skills and meet developmental milestones; however, after that point, they may stop developing new skills or lose previously acquired ones [3,9,10]. As ASD children grow into teenagers and young adults, they may struggle to make and keep friends, communicate with peers and adults, and understand the expectations for behavior in the classroom and at work [3,9]. Co-occurring disorders like anxiety, sadness, or attention-deficit/hyperactivity disorder, which are more prevalent in people with ASD than in those without ASD, may cause medical practitioners to identify them [3,6].

Individuals with ASD frequently struggle with interaction and social communication and demonstrate repetitive or restricted interests or actions. ASD subjects may also have unique learning, movement, and attention styles. These attributes can make life quite difficult.

### 1.1. Quantitative EEG in Autism Spectrum Disorder

Although the methods and experimental paradigms employed in these investigations differed, quantitative EEG (qEEG) has been used to evaluate autistic children, and it has revealed several consistently aberrant EEG characteristics in these individuals [11,12].

Quantitative EEG approaches may be able to identify and quantify ASD-related dysfunctions in certain brain regions and in the regulation of neuronal activity. Asymmetry of the hemispheres, functional connectivity, and spectral power have attracted the most attention as potential ASD EEG indicators. In their review of resting-state EEG investigations in ASD, Wang and colleagues [13] identified a probable “U-shaped” profile of EEG power spectra in ASD, in contrast to neurotypical control persons, with excess power in the theta and gamma frequency bands and decreased power in the alpha frequency band. Infants at risk for ASD exhibit alpha band imbalance between the hemispheres [14].

Cantor et al. [15] and Cantor and Chabot [16] demonstrated decreased alpha activity, while other publications reported bilateral and frontal decreases in this EEG frequency band [17] when employing spectral analysis. Children with autism exhibited significantly higher relative delta and lower relative alpha activity [18,19]. Additionally, these researchers noted that gamma oscillation alignment increases the sensitivity of distinguishing EEG responses to emotional facial stimuli in autistic individuals, and that an excess of high-frequency EEG demonstrates an association between autism and hyperactivity.

### 1.2. Retained Primitive Reflexes in Autism Spectrum Disorder

One early indicator of delayed or aberrant cortical maturation, and consequently, and other neurobehavioral disorder including ASD, may be retained primitive reflexes (RPRs) [20]. Sucking and rooting reflexes, as well as many other primitive reflexes, are present at birth [21]. The early indications of ASD may include delayed asymmetry of rolling over around 3–5 months of age, as well as the inability to latch on and breastfeed, which is frequently found in children with developmental delays [22,23]. Therapists recommend specific exercises that are thought to stimulate or reproduce primitive reflexes to remediate various neurobehavioral disorders [24,25].

However, to our knowledge, the mechanism by which primitive reflexes are suppressed in neurobehavioral disorders has not yet been established. We hypothesize that making use of these reflexes enhances sensory input and feedback to the nervous system, which in turn encourages synaptogenesis and neuroplasticity in more rostral and complex brain regions [26,27]. The suppression of primitive reflexes under normal circumstances is associated with inhibition through descending propriospinal connections. The suppression of these reflexes, which would result in associations with more intricate, individualized volitional movement control, promotes cortical maturation and growth. As a result, “bottom-up interference” may be released, delaying cortical maturation and preventing proper top-down regulation, both of which would ultimately block basic reflexes [21,28,29]. We have noted that in ASD individuals there is a significant tendency for primitive reflexes to be retained even into adult life [30].

The goal of the current study was to evaluate autistic children before and after an intervention, compared to a control group, using qEEG spectral and functional connectivity analysis, with an approach that can inhibit preserved basic reflexes [31].

## 2. Materials and Methods

### 2.1. Participants

Male and female participants were recruited from the Institute for Neurology and Neurosurgery in Havana, Cuba, tested and treated in the Clinical electrophysiology laboratory. Demographic data of the participants was recorded and included age, gender, Apgar score, birth weight, gestational age, and whether the birth was natural or by Caesarian section. Also collected were the IQ, overall health, grade level, and sidedness of the participants. Three age groups included 10 ASD participants each (5–10; 11–19; 25–35 years) reflecting different normative stages of development into adulthood. Additional selection criteria may be found in Section 2.1.1 and Section 2.1.2 below.

The participants included 50 males and 10 females whose mean age was 15.8 (S.D. 7.21). The groups’ characteristics can be found in a data depository at (https://www.researchgate.net/publication/372345066_Cognitive_Effects_of_Retained_Primitive_Reflex_Reduction_by_Lateralized_Stimulation-Data, accessed on 19 July 2023).

#### 2.1.1. Inclusion Criteria

Each participant was blindly clinically examined by two child neurologists and diagnosed as complaining ASD, based on DSM-V criteria [8].

Each of the ASD participants possessed a classical autistic triad of impairments in social interaction, communication, and imagination [7,32,33,34], with relatively intact verbal functions and with I.Q.s over 85 [35,36].

The following conditions were required for inclusion in the control group: a history of uneventful prenatal, perinatal, and neonatal periods; no disorders of consciousness; no history of central or peripheral nervous system disease, and the absence of the following: head injury with cerebral symptoms; convulsive episodes; paroxysmal EEG activity; headache; enuresis or encopresis after the fourth birthday; tics; stuttering; pavor-nocturnus; and any psychiatric, behavioral, or drug-related disorder. Depending on age, school-aged participants demonstrated normal academic achievement [8,34]. Control group participants were excluded if any spike-wave activity was present in the EEG.

#### 2.1.2. Exclusion Criteria

None of the participants had a history of cerebral palsy, traumatic brain injury (TBI), or brain surgery, nor did they have any neurologic abnormalities, outside those specifically linked to autism. None of the participants demonstrated any genetic disorder, metabolic illness, vascular disorder, or history of cancer, and they could not be breastfeeding or pregnant. Participants were free of drugs or in drug treatment recovery. Control group participants demonstrated no manifestations of any RPRs.

#### 2.1.3. Informed Consent and Institutional Approval

The Institute of Neurology and Neurosurgery Ethics Committee and the IRB for the University of Haifa approved the research projects. The relatives or responsible parties of the study participants gave their informed consent.

### 2.2. Procedures

Participants were studied in a facility with a temperature range of between 24 and 26 °C, with noise reduction, and dimmed lights. Participants under the age of majority had a parent or guardian in loco parentis present during all recording sessions, and in all cases, so too was the clinician in charge and the EEG technologist. To improve teamwork, participants were situated in comfortable chairs and familiarized with the room and experimental set prior to the experimental session.

#### 2.2.1. Reflex Testing

The following reflexes were tested clinically and included the Asymmetric Tonic Neck Reflex (ATNF), Symmetric Tonic Neck reflex (STNR), Spinal Galant, Babinski, Palmer Grasp, Rooting, and Tonic Neck Reflex (TNR). RPRs were defined as the presence of two or greater RPRs.

#### 2.2.2. Cognitive Testing

The following standardized testing was performed: Wechsler Individualized Achievement Testing-III (WIAT-III) [35], Wechsler Adult Intelligence Scale (WAIS-IV Spanish Edituon [37], Wechsler Intelligence Scale for Children Fourth Edition (WISC-IV, Spanish Edition) [38], Clinical Behavioral/Sensory Testing and Intervention, Behavioral Scales (Brown Executive Function Scales [39], Gilliam Asperger’s Disorder Scale (GADS) [40], and the Gilliam Asperger’s Rating Scale (GARS-3) [41]).

#### 2.2.3. Hearing and Vestibular Function

All children in Cuba with ASD are routinely tested by brainstem auditory evoked potentials to rule out auditory impairment. Cranial nerve function was also included as part of the evaluation protocol, as was the case in the patients evaluated in this report.

### 2.3. Quantitative Electroencephalography

EEG was recorded from 19 standard locations over the scalp according to the 10–20 system: Fp1, Fp2, F3, F4, F7, F8, T3, T4, C3, C4, P3, P4, T5, T6, O1, O2, Fz, Cz, and Pz. Gold-cup scalp electrodes applied with collodion were fixed, after careful skin cleaning, using a conductor paste and connected to the input box of the digital EEG system (Medicid-05, Neuronic, S.A., Neuronic Mexicana S.A. de C.V. Luis Alconedo #1, Colonia Merced Gomez, Delegación Benito Juarez, CDMX, México). Leads were employed, using linked ears as a reference. Technical parameters were: gain 20,000, pass-band filters 0.1–70 Hz, “notch” filter at 60 Hz, a noise level of 2 µV (mean root square), sampling frequency 200 Hz, and electrode–skin impedance never higher than 5 KΩ. To make eye movement artifacts easier to spot in the EEG records, electrodes were positioned across the superior and inferior rims. For each experimental portion, experts visually reviewed the recordings to choose artifact-free EEG segments with a total duration of no less than 65 s. These segments were afterwards converted to an ASCII file and saved for additional quantitative analysis.

#### 2.3.1. EEG Pre-Processing

Every single one of the 19 leads’ EEG values underwent off-line pre-processing consisting of (a) removing the sequence’s mean value from the EEG values to lessen the impact of the D.C. component of the time series; (b) applying a nonlinear median filter (three-points window) to weed out outliers or abnormally high amplitude values [42]; (c) conventional linear detrending to prevent any potential series drifts; (d) digital high pass filtering (low cutoff frequency of 0.5 Hz); and (e) lowpass digital filtering with a six order Butterworth filter and a high cutoff frequency of 55 Hz. An algorithm created by The Math Works Inc. was used for both filtering techniques. It filtered the input in the forward direction, reversed the filtered sequence, and then sent it back through the filter to create a zero-phase distortion effect [43].

#### 2.3.2. qEEG Spectral Analysis

qEEG samples contained in the previously specified ASCII files were imported by a custom-tailored Matlab version 7.10.0.499 R2010a software application (The Mathworks, Inc., 1 Apple Hill Drive, Natick, MA, USA). The software included several steps, including the estimation of the power spectral densities (PSD) for each EEG lead, the calculation of several spectral indices and coherence, and the output of these results to a Microsoft Access database.

#### 2.3.3. Grouping of EEG Leads for Spectral Analysis

Activity in the left anterior region was evaluated by recordings obtained through the EEG leads Fp1, F3, and F7. The right anterior area was composed of the Fp2, F4, and F8 derivatives. Activity over the central left region was evaluated by activity recorded from the C3 and T3 leads, and right central region activity was reflected in activity recorded from C4 and T4 leads. The posterior left region activity was reflected by the P3, O1, and T5 leads, and a posterior right region activity was integrated with the collective activity of P4, O2, and T6 leads; a midline region was defined that included the Fz, Cz, and Pz leads.

#### 2.3.4. Computation of PSD and the Spectral Indices

The first 12,288 samples of the EEG values of each EEG lead were submitted to a spectral analysis implemented by the Welch periodogram method, using a Hann window to avoid a leakage effect as much as possible. This approach produced 23 successive windows out of the 1024 samples (5.12 s), which overlapped every 512 samples. The estimated PSD findings for each discrete spectral frequency were then averaged to get the global smoothed spectrum for each EEG lead. This procedure had a spectral resolution of 1/5.12 s, or roughly 0.1953125 Hz. The D.C. or zero frequency and the first six discrete frequencies were eliminated, and the remaining discrete frequencies were subjected to integration within the ranges chosen for the various EEG bands. We selected 12 for the delta EEG band, which ranges from 1.17 to 3.5 Hz; 22 for theta, 3.5 to 7.5 Hz; 19 for alpha, 7.5 to 11 Hz; 21 for sigma, 11 to 15 Hz; 53 for alpha, 15 to 25 Hz; and 154 for gamma. The conventional method was also used to measure the PSD for the EEG in each band in normalized units, determining the percentage of the PSD corresponding to the total PSD of the entire spectral range [44,45].

#### 2.3.5. Functional Connectivity Examined by qEEG

The theoretical analysis of graphs was performed employing the timing matrix of all possible electrode combinations. We used programs developed in our laboratory employing MatLab (R2008b) for these analyses. The parameters that were evaluated were the mean route length (the smallest number of edges required to connect a node to another node) and the clustering coefficient (the ratio of connections between nearest neighbors to the total number of connections feasible), which both measure the effectiveness of communication in a network. The level of connectivity that the two neighbors or the local subnet share is reflected in the local efficiency. The global efficiency shows the degree of connectedness between any two nodes, and the global connectivity reflects the degree of global connectivity between each channel and the other metrics.

#### 2.3.6. Statistical Analysis

For the statistical processing, we used the Statistica v. 8.0. program (Tibco Inc., Palo Alto, CA, USA). By using log-natural transformations, the absolute PSD values were given a normal distribution, which was then verified using the Shapiro-W Wilk’s test. Based on our prior experience, normalized PSD data did not require adjustment to achieve normalcy distributions [46].

A permutation test implemented in Matlab was used to assess functional connectivity (FC), calculating the differences between the two groups’ synchronization matrices. Each matrix element in the FC symbol represents the edge strength or functional relationship between the associated groups, with the rows and columns standing in for nodes. The assembling of the element matrices into the global coefficient matrix is plainly shown by the connection matrix.

To analyze the differences between the controls and patients, the *t*-test for independent samples was applied. A significance of *p* < 0.05 was set in all cases.

## 3. Results

### 3.1. Effects of Treatment on Absolute Power

Comparing the control vs. the autism groups before the treatment, we found significant increments of the delta, theta, alpha, and gamma bands, with reduced activity of alpha frequencies.

Nonetheless, the most interesting finding was when ASD patients were compared before and after treatment, represented in Figure 1, Figure 2, Figure 3 and Figure 4. The most important findings related to comparisons prior to and after treatment, specifically a significant reduction of the absolute power in the delta, theta, and gamma bands. For the alpha band, no statistical differences were found.

### 3.2. Functional Connectivity

Figure 5, Figure 6, Figure 7, Figure 8 and Figure 9 compare qEEG functional connectivity before and after treatment for delta, theta, alpha, alpha, and gamma bands. We generally found long-range underconnectivity and short-range overconnectivity for all bands lateralized to the right hemisphere. Moreover, after treatment, several significant interhemispheric connections appeared.

## 4. Discussion

Comparing the control vs. the autism groups before the treatment, we found a significant increment of the delta, theta, alpha, and gamma bands, with reduced alpha frequency activity. This pattern is known as the “U Shape” in ASD patients. Nonetheless, the most important findings were related to the comparison before and after treatment.

The most common method for characterizing resting EEG involves decomposing oscillatory patterns into frequency bands with physiological features. Clinically important EEG frequency ranges typically span from 0.3 to 70 Hz. We concentrated on five frequency bands for the purposes of this paper: delta, theta, alpha, alpha, and gamma.

Clinical and cognitive neuroscience are becoming increasingly interested in these historically recognized frequency bands, which are considered to control a variety of brain activities. The event-related slow waves seen during activities that assess attention and salience are thought to be caused by delta [47]. Most often, theta research focuses on memory functions [48]. People who are relaxed and aware exhibit alpha waves, which are linked to the exact timing of sensory and cognitive inhibition [49]. Alpha waves are also associated with alertness, engagement in active tasks, and motor action [50]. Finally, gamma waves are hypotheszed to aid feature binding in sensory processing. They are present during working-memory matching and numerous early sensory responses [51]. In addition, slow waves are also associated with different lesions and functional dysfunction of the brain, which are outside the scope of this paper.

According to some experts, this U-shaped profile in ASD could be partially explained by aberrant gamma-aminobutyric acid (GABAergic) tone in inhibitory circuitry, which can affect the brain’s functional and developmental plasticity [13,52].

Dendritic GABAergic inhibitory dysfunction has been connected to gamma band activity observed in EEG seizure recordings [52,53,54]. However, elevating GABA concentration by giving the GABA antagonist vigabatrin to both rats and people has enhanced resting delta power [55,56], demonstrating that the U-shaped spectral profile in ASD cannot be fully explained by a simple decrease in GABA. In particular, GABAergic interneurons and glutamatergic neurons’ N-methyl-D-aspartate receptors, which are regulated by dopaminergic neurons in the thalamus, interact to produce thalamocortical delta oscillations. A complicated network of neurochemical changes that impact the physiology of inhibitory GABAergic interneurons and their control of excitatory activity in pyramidal cells may be the cause of the power anomalies in ASD. The so-called gamma binding has been related to cognitive processes [57].

During normal brain development, synaptogenesis allows for ongoing changes in both short- and long-range neural circuitry, weakening functional connections between nearby brain regions, while bolstering connections between various regions of the brain [58]. Developmental disorders like autism may alter the pathway, leading to abnormal brain connections. It is clear that ASD causes synaptic disruption both at the specific level of individual axons and the more general level of brain networks [59,60].

## 5. Conclusions

Researchers may be able to evaluate the changes in brain function between people with and without ASD by using qEEG coherence to look at electrical connection patterns. Our findings, particularly that of a high left-to-right qEEG functional connectivity ratio, indicate long-range underconnectivity and short-range overconnectivity with concurrent aberrant lateralization in ASD. Based on these results, we believe that ASD is characterized by a general trend toward the underexpression of low-band, widely distributed integrative processes, which is offset by localized, high-frequency, regionally specialized, and segregated processes.

Hence, clinical improvement and the disappearance of RPRs in our cases might be due to a new balance in qEEG frequency bands and a more optimized organization of the brain networks, improving long-range connectivities, mainly in the right hemisphere.

## Figures and Tables

**Figure 1 brainsci-13-01147-f001:**
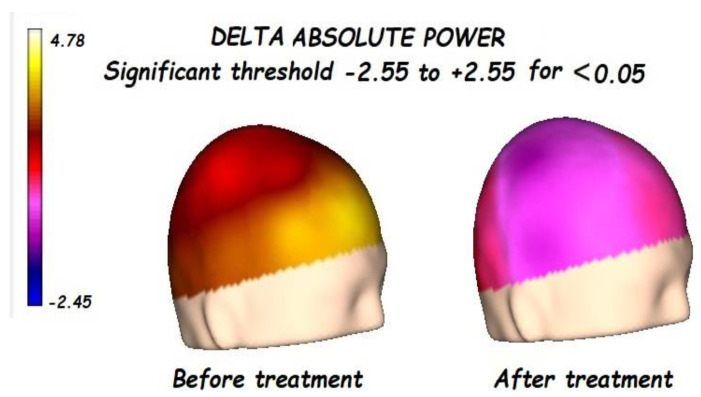
Before treatment, a significant global increment of delta absolute power was found, lateralized to the right hemisphere, and focalized in the frontal-temporal regions. After treatment, a significant reduction in the delta absolute power was found.

**Figure 2 brainsci-13-01147-f002:**
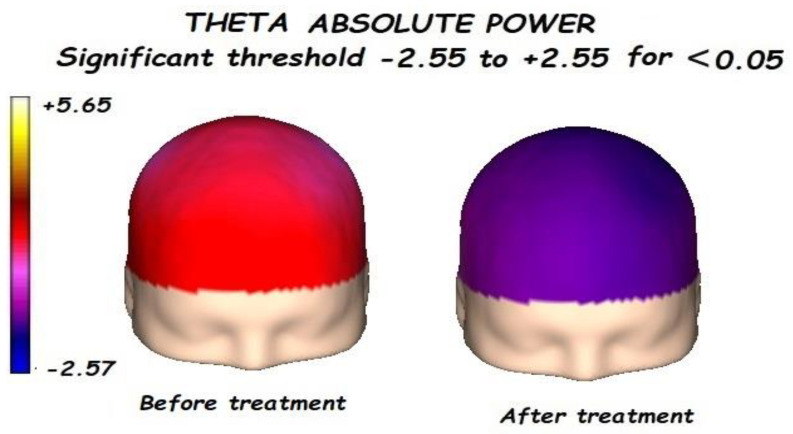
Before treatment, a significant global increment of the theta absolute power was found. After treatment, a significant reduction in the theta absolute power was found.

**Figure 3 brainsci-13-01147-f003:**
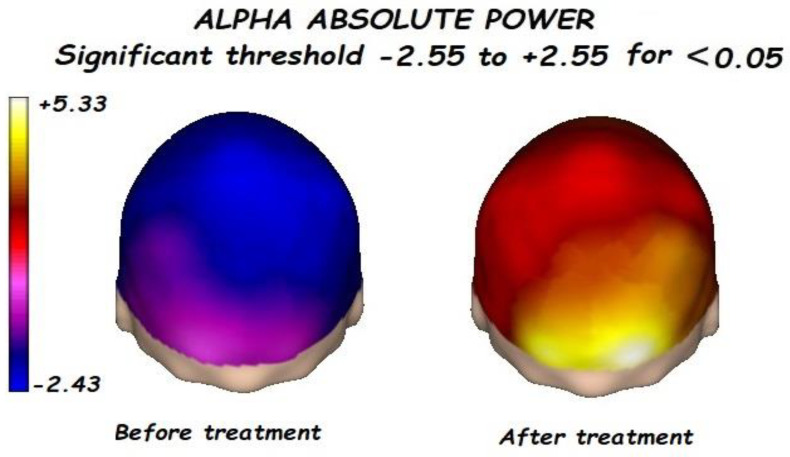
Before treatment, a significant posterior decrement of the alpha absolute power was found. alpha absolute power significantly increases after treatment.

**Figure 4 brainsci-13-01147-f004:**
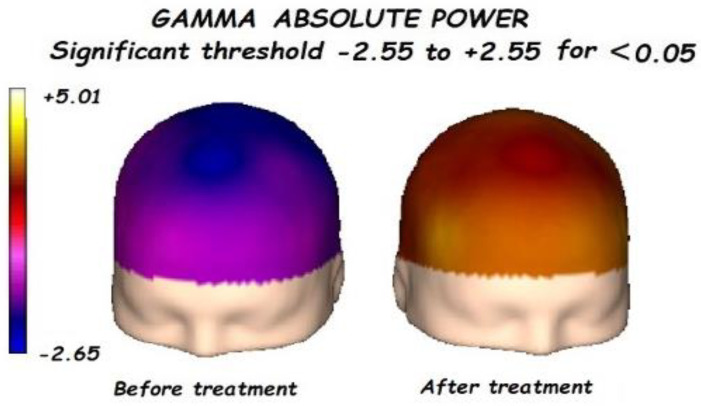
Before treatment, a significant global decrement in the gamma absolute power was found. Gamma absolute power significantly increases after treatment, mainly in the anterior regions.

**Figure 5 brainsci-13-01147-f005:**
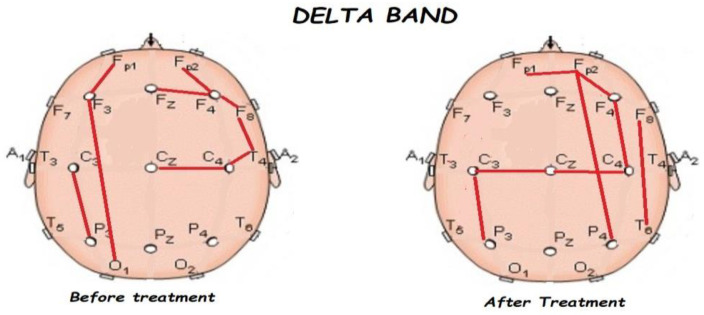
Comparison of functional connectivity before and after treatment for the delta band. Red lines indicate a significant connection in the network.

**Figure 6 brainsci-13-01147-f006:**
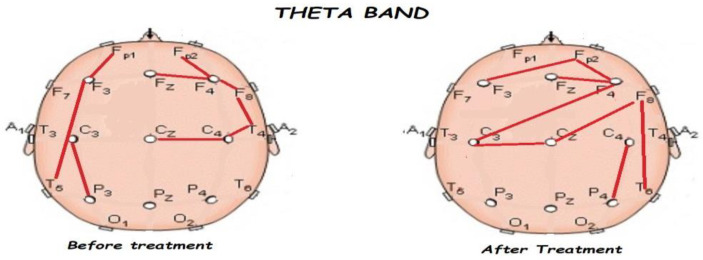
Comparison of functional connectivity before and after treatment for the theta band. Red lines indicate a significant connection in the network.

**Figure 7 brainsci-13-01147-f007:**
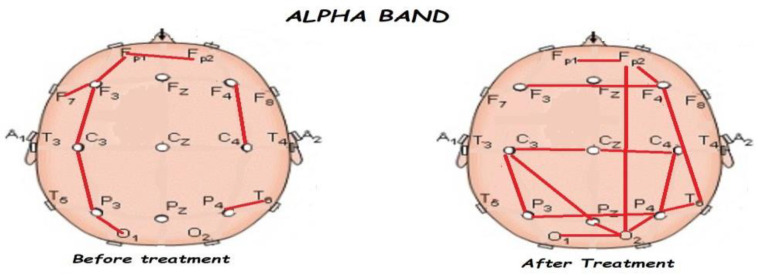
Comparison of functional connectivity before and after treatment for the alpha band. Red lines indicate a significant connection in the network.

**Figure 8 brainsci-13-01147-f008:**
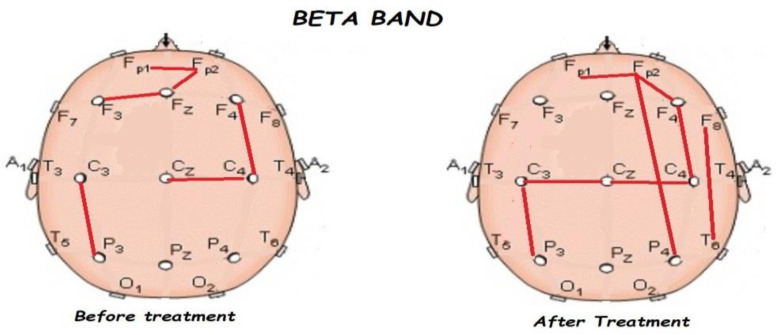
Comparison of functional connectivity before and after treatment for the beta band. Red lines indicate a significant connection in the network.

**Figure 9 brainsci-13-01147-f009:**
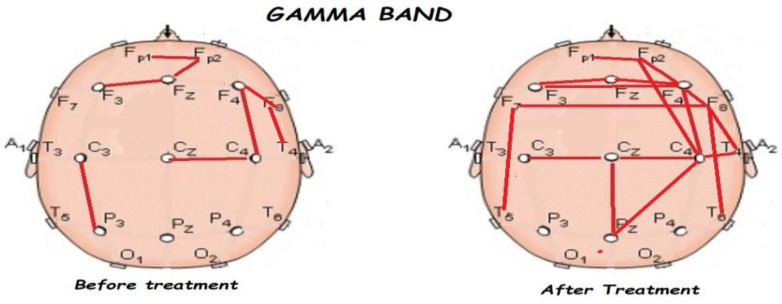
Comparison of functional connectivity before and after treatment, for the gamma band. Red lines indicate a significant connection in the network.

## Data Availability

Data supporting the results may be found at: https://www.researchgate.net/publication/372345066_Cognitive_Effects_of_Retained_Primitive_Reflex_Reduction_by_Lateralized_Stimulation-Data, accessed on 19 July 2023.

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
