# Peer review of "The Relationship between Retained Primitive Reflexes and Hemispheric Connectivity in Autism Spectrum Disorders"

_brainsci, 2023, doi:10.3390/brainsci13081147_

Round 1
Reviewer 1 Report
Comments and Suggestions for Authors
The article presents the EEG findings for establishing a Relationship between Retained Primitive Reflexes and Hemispheric Connectivity in Autism Spectrum Disorder.
I appreciate the opportunity to review the manuscript submitted to brainsci-2500554. I have found an interesting work providing innovative research questions, and attending to health demands.
However, I have found several issues that must be attended by the authors .
1. Introduction. Lines 32 to 54 require to be referenced. Authors provide relevant and interesting affirmation, but without referencing. 2. Methods: lines 137-138, it was the technician who assessed all variables afterward? if so, please provide information regarding the experience of the persons in charge to make the evaluations and recordings. Section 2.2.2 Please provide references for the implemented scales, it would be relevant to inform if scales were properly standardized for the population.
The participants' section should precise the total number of participants, and how groups were assigned, what was the reason for exploring the variables in these age ranges?
Line 207 alpha band is repeated and explained out of range. 2.1.1 Statistical analysis, please provide further details about groups matrices synchronization analysis, it is unclear if the authors refer to matrices as the result of connectivity previously explained procedure.
The authors used a t-test for comparing controls and participants, however, the objective explained about an intervention program, if they did so, then there are four groups of data, leading to the need of performing an ANOVA test instead, only if data distributed normally.
3. Results
Lines 232-234, It is not clear which groups the authors are referring to, this leads to a quite severe problem, once they previously mentioned an adults 25-35 years old group, where no issues in alpha band are expected, even in the 11-19 group.
In my particular opinion, at this point, the authors require to deliver more details specifically regarding groups of data, for improving the interpretation of the results.
General comments, some initials need previous definitions, and the same format (for example qEEG instead of QEEG). Typo in line 66. APGAR score (line 104) is initials. English style needs to be corrected. The numerical sequence in procedures needs to be corrected.
English Language style needs revision.
Author Response
1. The article presents the EEG findings for establishing a Relationship between Retained Primitive Reflexes and Hemispheric Connectivity in Autism Spectrum Disorder. I appreciate the opportunity to review the manuscript submitted to brainsci-2500554. I have found an interesting work providing innovative research questions, and attending to health demands. However, I have found several issues that must be attended by the authors .
1A. We thank the reviewer for the encouraging words.
2. Introduction. Lines 32 to 54 require to be referenced. Authors provide relevant and interesting affirmation, but without referencing.
2A. This has been done and we apologize for the oversight
3. Methods: lines 137-138, it was the technician who assessed all variables afterward? if so, please provide information regarding the experience of the persons in charge to make the evaluations and recordings.
3A. Actually, not so. The technician of course was present with the parents, but assessments were made by a license physician. For better clarity, we have reworded the sentence thusly, “Participants under the age of majority had a parent or guardian in loco parentis, present during all recording sessions, and in all cases, so too was the clinician in charge and the EEG technologist.”
4. Section 2.2.2 Please provide references for the implemented scales, it would be relevant to inform if scales were properly standardized for the population.
4A. All tests have been standardized and appropriate references supplied in section 2.2.2.
5. The participants' section should precise the total number of participants, and how groups were assigned, what was the reason for exploring the variables in these age ranges.
5A. Whoops! Thank you for pointing that out. The participants have now been described along with the selection characteristics and link to the data repository has been added to the paper in section 2.2.2.
6. Line 207 alpha band is repeated and explained out of range.
6A. In reviewing the paper, we found that the reviewer’s careful identified an error of fact. There exists a) a significant reduction in delta and theta, b) the theta illustration was repeated twice, and we have now replaced that erroneous figure with the correct one and c) a significant increase in both gamma AND alpha. The error has been corrected and is now represented between line 244-246. Additionally, this likewise correct the “very big problem” in item 8 below.
7. 1.1 Statistical analysis, please provide further details about groups matrices synchronization analysis, it is unclear if the authors refer to matrices as the result of connectivity previously explained procedure.
7A. The text now reads, “A permutation test implemented in Matlab was used to assess functional connectivity (FC), calculating the differences between the two groups' synchronization matrices. FC is denoted as a matrix with the rows and columns representing nodes and each matrix element representing the edge strength or functional connection between the corresponding groups. The connection matrix clearly illustrates the assembly of the element matrices into the global coefficient matrix.”
8. The authors used a t-test for comparing controls and participants, however, the objective explained about an intervention program, if they did so, then there are four groups of data, leading to the need of performing an ANOVA test instead, only if data distributed normally.
8A. The point is understood, but the only thing that is being examined in the instant paper are differences in frequency bands pre/post treatment and connectographic differences that are treated separately. So, Student’s t would be a simple but appropriate measure for examining pre-post differences only. The cognitive effects and their interaction have not been examined here.
9. Results. Lines 232-234, It is not clear which groups the authors are referring to, this leads to a quite severe problem, once they previously mentioned an adults 25-35 years old group, where no issues in alpha band are expected, even in the 11-19 group.
9A. We thank the reviewer for pointing out the egregious embarrassing error in not having proofread the manuscript thoroughly. We have corrected the problem and the explanation can be found in the answer to item 5 above. We had not noticed that the theta was inserted twice instead of the alpha band data. Also, the alpha data indicates an increase rather than the opposite.
10. In my particular opinion, at this point, the authors require to deliver more details specifically regarding groups of data, for improving the interpretation of the results.
10. We have resolved the issue, in part, my having created a link to a data repository that has all of the aggregated data from the project with the appropriate headings the link is now included in the body of the paper.
11. General comments, some initials need previous definitions, and the same format (for example qEEG instead of QEEG). Typo in line 66. APGAR score (line 104) is initials. English style needs to be corrected. The numerical sequence in procedures needs to be corrected.
11. We have reviewed the paper now numerous times for spelling, grammatical errors, and style and hopefully it will now read better.
Reviewer 2 Report
Comments and Suggestions for Authors
Dear authors,
An interesting article for the scientific community on primitive reflexes in people with Autism Spectrum Disorder is presented. The references provided are key to understanding the work done.
However, I also present some suggestions for improvement.
- The title reflects the topic well.
- Keywords are clear and concise.
- The abstract could be improved. At the beginning it is written that the study has evaluated children and adults with ASD, but it would be appropriate to start by talking about ASD, primitive reflexes or hemispheric connectivity to better understand the topic with a quick glance at the abstract.
- There are many authors who detail the age at which approximately the ASD begins to manifest itself. Said articles should be cited in lines 36-41.
- In lines 51-52 it would be interesting to quote the DSM 5 TR (2022)
- In the introduction section it is strongly recommended to modify the cited articles. These are very old and barely correspond to the current research regarding the last five years (2018-2023).
- The materials and methods have been well described.
- Normally the approval number of the Ethics Committee of the Institute of Neurology and Neurosurgery is indicated. Therefore, it is recommended to indicate it.
- The subscripts are not well numbered. Numbers 2.2.1 are constantly repeated. This must be corrected.
- The figures are necessary to better understand the results section.
- The discussion section presents very well the results found and their connection with current previous studies.
- A conclusions section should be added where the main findings are summarized and a reflection is made on them. In addition, it is important to add the limitations of the study and the prospective research.
Author Response
1. An interesting article for the scientific community on primitive reflexes in people with Autism Spectrum Disorder is presented. The references provided are key to understanding the work done. However, I also present some suggestions for improvement. The title reflects the topic well. Keywords are clear and concise.
1A. We thank the reviewer for the kind comments. The rest we will deal with in seriatim
2. The abstract could be improved. At the beginning it is written that the study has evaluated children and adults with ASD, but it would be appropriate to start by talking about ASD, primitive reflexes or hemispheric connectivity to better understand the topic with a quick glance at the abstract.
2A. The abstract has been rewritten accordingly.
3. There are many authors who detail the age at which approximately the ASD begins to manifest itself. Said articles should be cited in lines 36-41.
3A. Citations have been added accordingly.
4. In lines 51-52 it would be interesting to quote the DSM 5 TR (2022)
4A. This has been done
5. In the introduction section it is strongly recommended to modify the cited articles. These are very old and barely correspond to the current research regarding the last five years (2018-2023).
5A. The recommendations are accepted and have been implemented.
6. The materials and methods have been well described.
6A. We thank the reviewer for the positive vote.
7. Normally the approval number of the Ethics Committee of the Institute of Neurology and Neurosurgery is indicated. Therefore, it is recommended to indicate it.
7A. We have added the approval number of the INN’s ethics panel.
8. The subscripts are not well numbered. Numbers 2.2.1 are constantly repeated. This must be corrected.
8A. The subscripts have been appropriately adjusted.
9. The figures are necessary to better understand the results section.
9A. The figures had been inserted into the template. They have been reworked for better clarity
10. The discussion section presents very well the results found and their connection with current previous studies.
10A. We thank the reviewer for the positive comment.
11. A conclusions section should be added where the main findings are summarized and a reflection is made on them. In addition, it is important to add the limitations of the study and the prospective research.
11A. This has been done.